

**Projections of the Affluent Natural Energy (ANE) for the**
**Brazilian electricity sector based on RCP 4.5 and RCP 8.5**
**scenarios of IPCC-AR5**
Cleiton da Silva Silveira[1], Francisco de Assis de Souza Filho[2], Francisco das Chagas Vasconcelos
Junior[2,3] Eduardo Sávio Passos Rodrigues Martins[3]
[1]University of International Integration of the Afro-Brazilian Lusophony (UNILAB), Redenção, Brazil
[2]Department of Hydraulic and Environmental Engineering, Federal Ceará University, Campus do Pici, Fortaleza, Brazil
[3]Fundação Cearense de Meteorologia e Recursos Hídricos (FUNCEME), Aldeota, Fortaleza, Brazil
*Correspondence to:* Cleiton da S. Silveira (cleitonsilveira@unilab.edu.br)
**Abstract**. The Affluent Natural Energy (NAE) of Brazilian hydroelectric exploitations that comprise the National
Interconnected System (NIS) was obtained from the projections of global IPCC-AR5 models for RCP 4.5 and RCP 8.5
scenarios. The analysis considered the periods from 2010 to 2039, 2040 to 2069 and 2070 to 2098, under RCP4.5 and
RCP8.5 scenarios, in relation to 1950-1999. Streamflows were generated for 21 basins of the National Interconnected
System (NIS) by using the hydrological model Soil Moisture Account Procedure (SMAP). The model was initialized by
bias-removed monthly precipitation from the Climatic Research Unit (CRU) and estimated potential evapotranspiration
using Penman-Monteith method. Streamflows in the other 182 stations of the NIS were obtained using linear regression
with monthly streamflow computed by SMAP as predicted variables. NAE was obtained through product between the
naturalized streamflow into each station and the productivity from each hydroelectric exploitation. The models
projections indicate that the Brazilian hydropower system may suffer reductions in mean annual streamflow on most
basins. This decreasing also suggests a reduction in the mean annual NAE values in the NIS. However, on Southern
subsystem most models projected increase of annual NAE. This information defines the bounds for potential future
streamflow scenarios and it can be used for the adoption of management policies.
**Keywords:** Brazilian Hydropower Sector, IPCC-AR5, Climate Change, Streamflow, Penman-Monteith, Affluent
Natural Energy.





## 1 Introduction

Climate change can produce large impacts on water resources. The observed warming during the recent decades can cause changes in the hydrological cycle due changes in precipitation and evaporation spatial-temporal patterns. The changes directly affect soil moisture, groundwater reserves and the streamflow seasonality. These aspects associated with projected increases of water demand for the coming decades, primarily due to population growth and increasing wealth - regionally - may put the Brazilian hydroelectric system under pressure (Silveira et al. 2014).

The average annual streamflow in the Brazil Rivers is 179 thousands m3/s, which correspond to approximately 12% of the surface water availability on Earth (Shiklomanov 1998). The rivers streamflow variation is influenced by several factors, including the precipitation occurred in the contribution basin and land use/occupation changes. In Brazil the average annual rainfall (history from 1961 to 2007) is 1,761 mm, varying values in the range of 500 mm in the semiarid region of the Northeast and more than 3,000 mm in the Amazon region (ANA 2009).

The Brazilian power grid is most characterized by hydroelectric sources. The hegemony of hydropower in electricity production in Brazil requires careful analysis of streamflow into hydrological system and its temporal variation patterns due to the significant impact from these variations may have on the energy supply, and consequently on the entire national economy (Alves et al. 2013). In addition to natural climate variability, some studies suggest that climate change may affect weather and hydrological variables in South America (Silveira et al. 2014; IPCC 2007; Nobre 2005; Silveira et al. 2013A; Silveira et al. 2012; Marengo & Soares et al. 2005; Marengo & Valverde 2007).

The Intergovernmental Panel on Climate Change (IPCC) points out that climate change poses a major threat to socio-economic development during the 21st century, their impacts will likely affect in global scale the water resources, urban and rural infrastructure, coastal areas, forests and biodiversity, as well as economic sectors such as agriculture, fishing, forestry and power generation industries (IPCC 2014b).

In South America, climate variability and climate changes have been the subject of discussions and scientific research seeking a comprehensive understanding of their occurrence and characteristics (Chaves et al. 2004; Nobre 2005; Silveira et al. 2013; Silveira et al. 2012; Marengo & Soares et al. 2005; Marengo & Valverde 2007). Based on projections of global models, the IPCC has indicated that there is a higher probability of the maintenance of continuous heating throughout South America, as well as decrease in precipitation patterns over the Andes with an increase in the far south and south-east. In northern and northeastern South America, there is a reduction of





precipitation pattern during austral summer, but great divergence among the models projections (IPCC 2014a).

The impacts of climate change in the surface water flow and groundwater recharge varies depending on region and climate scenario considered (IPCC, 2014b), but relates mostly with changes designed for precipitation patterns (IPCC, 2001; KROL et al., 2006).

Several studies were conducted in order to check climate change impact on water resources in South America. Milly et al. (2005) showed that there is concordance on projections to the middle of the 21st century showing increased flow on the basin of the Paraná-Prata and reduction in the basins of eastern Amazonia and the Northeast of Brazil based on IPCC-AR4 models in the A1B scenario. Tomasella et al. (2009) have engaged a hydrological model for large basins to a regional climate model forced with global model HadCM3 A1B scenario of the IPCC AR4-and this showed 30% reduction in the monthly flow projections on the Tocantins River basin, indicating that further reductions may occur during the dry season. Silveira et al. (2014) analyzed the impact on streamflows in the basins of the Brazilian electric system based on projections of the IPCC-AR4 global models, forcing hydrological rainfall-runoff models for the A1B, A2 and B2. The results show wide divergence in the annual flow values for the Northeast of Brazil, especially in the basin of Xingó, together with associated reductions in flow rates in the Southeast sector of the country, around 5% in Furnas. Lucena et al. (2010) highlighted that Northern and Northeastern Brazil have vulnerabilities assigned to climate changes and suggest major impacts on hydropower grid if the projections from IPCC AR4 models were to be achieved. Even with the effects from the impact caused on the land and water ecosystems in mind, some studies (Straskraba & Tundisi 1999; Tundise 2007) suggested that benefits from construction of reservoirs could arise along of time. However, recently, Prado Jr. et al. (2016) indicate the expansion of hydropower generation designed for the northern sector of the country can lead huge socioeconomic and environmental impacts in Brazil, and projects the needs for increasing the participation of other renewable energy sources in power grid of the country.

A measure that defines the potential from power generation system is a hydroelectric Affluent Natural Energy (ANE), which is based on the average productivity of the system according to the streamflow into each reservoir. The objective of this study is to evaluate the projections of ANE for the National Interconnected System (NIS) based on scenarios RCP4.5 and RCP8.5 from IPCC-AR5 models for three time-slices, 2010-2039, 2040-2069, and 2070-2098.





## 2 The National Interconnected System (NIS)

The National Interconnected System (NIS) is responsible for production and transmission of electric energy in Brazil. It is a large hydrothermal system with predominance of hydroelectric plants, where only 3.4% of the country's electricity production capacity are out of SIN, in small isolated systems located mainly in the Amazon region (ONS, 2011a).

In order to make the most of existing NIS energy resources and the very hydrological seasonality of each region the system is divided into four subsystems: southeast/Midwest, southern, north and northeast industry sector. These subsystems are interconnected by an extensive transmission network that enables the transfer of energy surpluses and allows the optimization of inventory stored in the reservoirs of hydroelectric plants.

The Northern region produces 8% of the national energy (in GWh) and demand 7%, the Northeast region produces 11% and demand 18%, the South produces 18% and demand 16%, the Southeast/Midwest produces 47% and demand 61%. As for inter-regional transfers the North transmits 31.7% of its production to the Northeast; the Southeast/Midwest transmits 3.5% and 2.4% of its production to the North and Northeast, respectively, and receive 6.7% of production in the South and 100% of the Itaipú plant production to Brazil.

In Figure 1 is shown the ANE for the four NIS subsystems. This indicates an existing hydrological complementarity between sectors Southeast /Midwest and South. The distribution of monthly average affluent natural energy shows that their wet and dry periods do not coincide in these regions. The spatial structure of the NIS causes there is a spatial and temporal coupling of the decisions taken in its energy operation.

## 3 Methodology

### 3.1 IPCC-AR5 Models

Monthly data from the IPCC-AR5 global models simulations are considered in this work. For this experiment, just models that provided the maximum and minimum air temperature were used, because the estimation method of the potential evapotranspiration suggested in this work. Several models from research centers worldwide that contributed to the AR5 report are according to Table 1. These models were forced by observed concentrations of greenhouse gases during the 20th Century (Historical Simulations). Additionally, sets of projection simulations with emission scenarios of these gases (Representative Concentration Pathways, RCP) over the 21st Century were



also considered (RCP simulations). These scenarios are related to total radiative forcing in 2100. This work analyzes the RCP4.5 and RCP8.5 projections, which assume an increase in radiative forcing of 4.5 and 8.5 W/m² by the end of century relative to pre-industrial period.

## 3.2    Scenarios for the 21$^{st}$ Century

The scenarios RCP4.5 and RCP8.5 are considered for the evaluation of the ENA projections over the 21$^{st}$ century for the National Interconnected System. 10 models were considered from RCP4.5 simulations, whereas for RCP8.5 projections 14 models were utilized.

The scenario RCP4.5 assumes that forcing stabilizes shortly after 2100 without overshooting the long-run radiative level of 4.5 W/m$^2$ (Clarke et al. 2007). These projections are consistent with low energy intensity, strong reforestation programs and decreasing use of agricultural lands, stringent climate policies, stabilization of methane and $CO_2$ emissions reaching the target value of 650 ppm $CO_2$ equivalent during the second half of the 21$^{st}$ century (Vuuren et al. 2011).

The scenario RCP8.5 suggests a continuous growth of the world population associated with slow technological development, resulting in accentuated carbon dioxide emissions. This scenario is considered the most pessimistic for the 21$^{st}$ century in terms of greenhouse gas emissions, being consistent with no policy change to reduce emissions, rapid increase in methane emissions and heavy reliance on fossil fuels (Riahi et al. 2007).

### 3.3 Streamflow data

The methodology for getting estimated streamflows is divided basically into five stages:

● The first step is to select the representative basins that contemplate spatially most of the NIS, by selecting 21 basins, arranged as in Figure 2. Then are selected INMET data stations in the period 1992-2007 (precipitation, solar radiation, temperature, humidity) for calibration parameters of the SMAP hydrological model. Next we used an objective optimization procedure based on the Nash-Shutcliffe coefficient by comparing the number of streamflow obtained by the SMAP with the series provided by ONS (Silveira et al. 2014);

● The second step consists in obtaining the precipitation of global IPCC-AR5 models for the Historical scenarios, RCP 4.5 and RCP 8.5 to 21 basins of interest for three time-slices 2010-2039, 2040-2069, and 2070-2098, as shown in Figure 2, for statistical correction (bias removal) by using the Gamma distribution function. This correction has utilized monthly precipitation of the Climate Research Unit data base (CRU) (New et al. 2001).   This dataset corresponds to monthly precipitation from 1950 and 1999 with a spatial resolution of 0.5x0.5 degrees for land only (CRU





TS 3.0). For historical period was used the monthly series from 1950 to 1999, while for the
projections was used the series from 2010 to 2098.
● Third step is to get the potential evapotranspiration from the global IPCC-AR5 models for the
Historical scenarios (RCP 4.5 and RCP 8.5) using the Penman-Monteith method (Allen et al. 1998).
To do that, we utilized as input field data the maximum, minimum and average temperature time-
series from the global IPCC-AR5 models.
● In the fourth step we obtain the flows using the Soil Moisture Accounting Procedure - SMAP
(Lopes et al 1981; Souza Filho & Porto, 2003) - in 21 stations, taking as input the estimated
evapotranspiration and precipitation with bias removed.
● In the fifth step we obtain the flows estimated from the data of global models for stations that do
not have the hydrological model calibrated, through monthly regressions from data stations that
have the SMAP model calibrated. To perform this procedure are used the monthly naturalized flows
series provided by ONS in period 1931 to 2009. The ONS flows series are divided into two groups:
one containing 21 and other containing 167 stations. The monthly flow of the 21 stations was used
as predictors of other stations. So we obtain the parameters of the 167 stations using stepwise
regression.
Streamflows from the global IPCC-AR5 models were estimated by the SMAP for the 21
basins and for another 167 stations from NIS by using linear regression due strong relation among
the stations.
**3.4 Affluent Natural Energy Calculation (ANE)**

The natural streamflow are estimated for each hydroelectric power plant following the schematic
diagram shown (ONS, 2011b). The ANE is calculated from the natural streamflow and the
productivity equivalent to 65% of useful storage volume of reservoirs of the hydroelectric
exploitations (ONS, 2010). The values of ANE can be calculated on a daily, weekly, monthly, or
yearly scale and also by basin and by subsystem, according to the existing hydroelectric
exploitations systems in watersheds and electrical subsystems. The equation that defines ENA is the
follow expression:


$$ENA_{subsistema}(t) = \sum_{i=1}^{n}[Q_{nat}(j,t).p(j)] \qquad (1)$$





where
t= ANE time step calculation;
i= Exploitations (subsystem considered);
n= Number of exploitations (subsystem considered);
$Q_{nat}$ = Natural flow of the exploitation in the considered time step;
p = average productivity of the turbine-generator set relating to the drop obtained by the difference
between the upstream level corresponding to a storage 65% of the useful volume and the average
level of the tailrace.
**4 Results**

12         The medians of annual mean streamflow anomalies from global models projections indicate

that the impact on Brazil is not uniform throughout Brazil. It is more likely the occurrence of
moister conditions over southernmost Brazil at the end of 21st century, while other regions of the
country will probably present reductions in streamflow according to Figure 3.

16         Northern sector there is a fair spatial spreading streamflow response from climate change

scenarios. Stations northernmost show lower reduction compared to southernmost stations. In long-
term period (2070-2098) for RCP8.5 scenario (Figure 2f), for example, the reduction in anomalies
are between -5% and 0% northernmost whilst -15% and -20% southernmost.

20         Southeastern/Center-Western sector also presents slight spatial divergences in annual

streamflow anomalies. The median of anomalies shows two regions that have different response
from the climate change scenarios, centermost part of the Brazil and coastal area of Southeast
region. This characteristic is highlighted in RCP4.5 scenario during period from 2040 to 2069
(Figure 3c), the median on the first region several points with reduction between -10% and -15%,
while on the second the values are found between -5% and 5%.

26         In northeastern sector the median of anomalies in Sobradinho and Xingó basins reaches

values below -20 %. We can see that from RCP4.5 to RCP8.5 there is a decreasing in median of
anomalies (Figure 3d) during all period, this indicating the semiarid regions are very sensitive to
increasing greenhouse concentration in the atmosphere.

31         Most models show a reduction in annual ANE sectors in North, Northeast and Southeast in

the three periods and for both scenarios, as shown in Figure 4. For the South subsystem most
models show an increase in the three periods.

34         The global models for RCP 8.5 scenery indicates projections with more intense ANE

reductions than the RCP 4.5 for the North, North East and Southeast subsystems. This may be





associated with a higher temperature increase 8.5 designed by RCP, and consequently increased
evapotranspiration potential designed for the twenty-first century which directly impacts the
naturalized flow.
Furthermore, the positive anomalies shown to the South subsystem are more intense for the
scenario RCP 8.5 than for the RCP 4.5, indicating the possibility of years with heavy rains in the
first scenario and most runoff.
In Table 2 are shown the trend of the slopes of global models for RCP 4.5 and RCP 8.5
scenarios for ANE Brazilian subsystems using the Man-Kendall Sen test of annual flows. Analyzing
the set of models that have significant trend considering all the NIS, there is clear evidence that
increased emissions of greenhouse gases suggests a greater impact on power sector power
generation, since in most cases the slope module is always higher for the RCP 8.5 scenario than for
RCP 4.5.
In the RCP 8.5 scenario and for the North and North East subsystems, most models indicates
negative trend, being up to -6.1% per decade for the North and -3.9% per decade for the Northeast.
In addition, six models showed a positive trend for the South subsystem, with very expressive
magnitudes, with models indicating up to 13.2% per decade. While in the Southeast sector no
significant trends are found, five models indicated negative trend, with the MIROC-ESM model
showing -6.2% per decade.
In the RCP 8.5 scenario and for the North and North East subsystems, most models indicates
negative trend, being up to -6.1% per decade for the North and -3.9% per decade for the Northeast.
In addition, six models showed a positive trend for the South subsystem, with very expressive
magnitudes, with models indicating up to 13.2% per decade.
The CSIRO-Mk3-6-0 and CanESM2 models indicate negative trend in Southeast, Northeast
and North subsystems and no trend in the Southern sector for both scenarios of the IPCC-AR5. This
configuration results in negative trend for the NIS up to -3 3% per decade, according to CanESM2
model for RCP 8.5 scenario.

**5 Conclusions**


The climate has a strong influence on the development of society due to floods, droughts,
disasters, among many other factors that directly affect the environment, agriculture, energy sector,
air quality etc. The analysis proposed in this paper aimed to provide information on the impact of
climate change on Energy Affluent Natural and consequently in energy production in Brazil. This



information can be used by managers to adopt energy policies and assist actions to minimize the
impacts of such scenarios.

3       Most models indicate that the Brazilian electric sector may decrease in ANE. Since this this

decrease is not spatially linear, most models suggest that the South subsystem increases the energy
produced, together with substantial reductions of ANE in sectors North, Northeast and Southeast
compared to the historical period.
The possible reduction in ANE indicated by most models suggest that climate change added to the
growth of energy demand in Brazil could lead to a major crisis in the Brazilian energy sector,
leading to investments in non-renewable energy because of the risk of non-compliance of users with
current energy matrix. This type of action may increase the cost of energy generation and create a
positive feedback to climate change and intensify their effects on the entire climate system, and
consequently on the country.
The other possibility front of projected climate change is a massive investment in renewable
sources such as wind and solar, so that they can reach a large stake in Brazilian power matrix.
However, this requires a complex policy of investment in technology and labor-intensive training
for the long-term cost of generation become cheaper for the country.

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





Figures and Tables

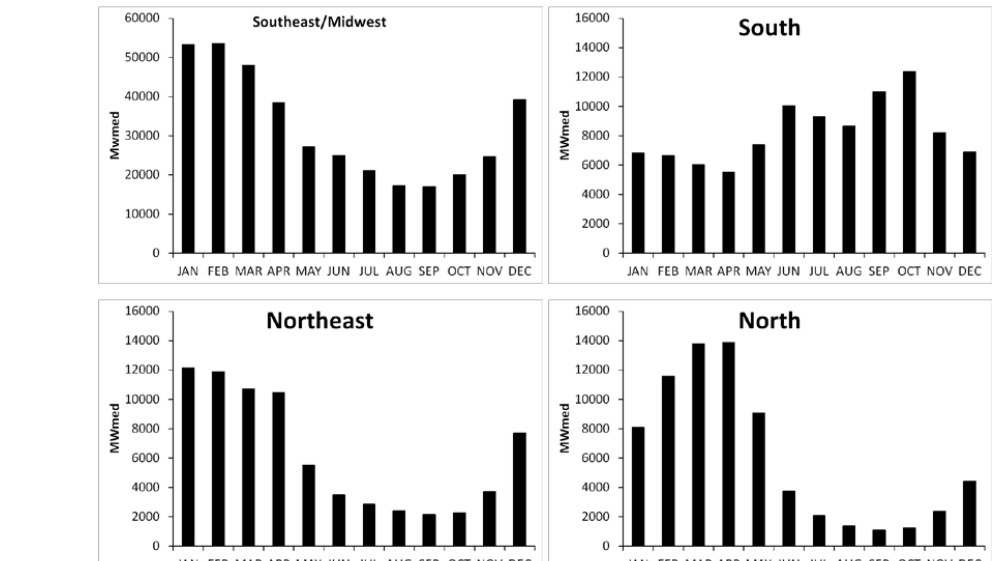

Figure 1 - ANE in MWmed (the unit used is the relationship between the generated power and the operating time of the
facility) of the Brazilian electricity sector subsystems from 2000 to 2014. Data source: ONS.








2                       Table 1 – Global Models of the IPCC-AR5 considered.

| Model | Institution or Agency; Country | Scenarios |
|---|---|---|
| bcc-csm1-1 | *Beijing Climate Center, China Meteorological Administration; China* | *RCP 4.5 and RCP8.5* |
| BNU-ESM | *College of Global Change and Earth System Science, Beijing Normal University; China* | *RCP 4.5 and RCP8.5* |
| CanESM2 | *Canadian Centre for Climate Modelling and Analysis; Canada* | *RCP 4.5 and RCP8.5* |
| CESM1-BGC | *Community Earth System Model Contributors, USA* | *RCP 4.5 and RCP8.5* |
| CSIRO-Mk3-6-0 | *Commonwealth Scientific and Industrial Research Organization in collaboration with Queensland Climate Change Centre of Excellence; Australia* | *RCP 4.5 and RCP8.5* |
| GFDL-ESM2M | *NOAA Geophysical Fluid Dynamics Laboratory; USA* | *RCP 4.5* |
| HadGEM2-AO | *National Institute of Meteorological Research/Korea Meteorological Administration; UK* | *RCP 4.5 and RCP8.5* |
| HadGEM2-CC | *Met Office Hadley Centre; UK* | *RCP 4.5* |
| HadGEM2-ES | *Met Office Hadley Centre; UK* | *RCP 4.5* |
| IPSL-CM5A-LR | *Institut Pierre-Simon Laplace; France* | *RCP 4.5 and RCP8.5* |
| IPSL-CM5A-MR | *Institut Pierre-Simon Laplace; France* | *RCP 4.5 and RCP8.5* |
| MIROC5 | *Atmosphere and Ocean Research Institute, National Institute for Environmental Studies, and Japan Agency for Marine-Earth Science and Technology; Japan* | *RCP 4.5 and RCP8.5* |
| MIROC-ESM | *Atmosphere and Ocean Research Institute, National Institute for Environmental Studies, and Japan Agency for Marine-Earth Science and* | *RCP 4.5 and RCP8.5* |





*Technology; Japan*



2    Figure 2 - Hydroelectric exploitations of the NIS. The square represents the stations that were generated from the use of
3    the hydrological model, while the "x" represents the stations generated from linear regressions.





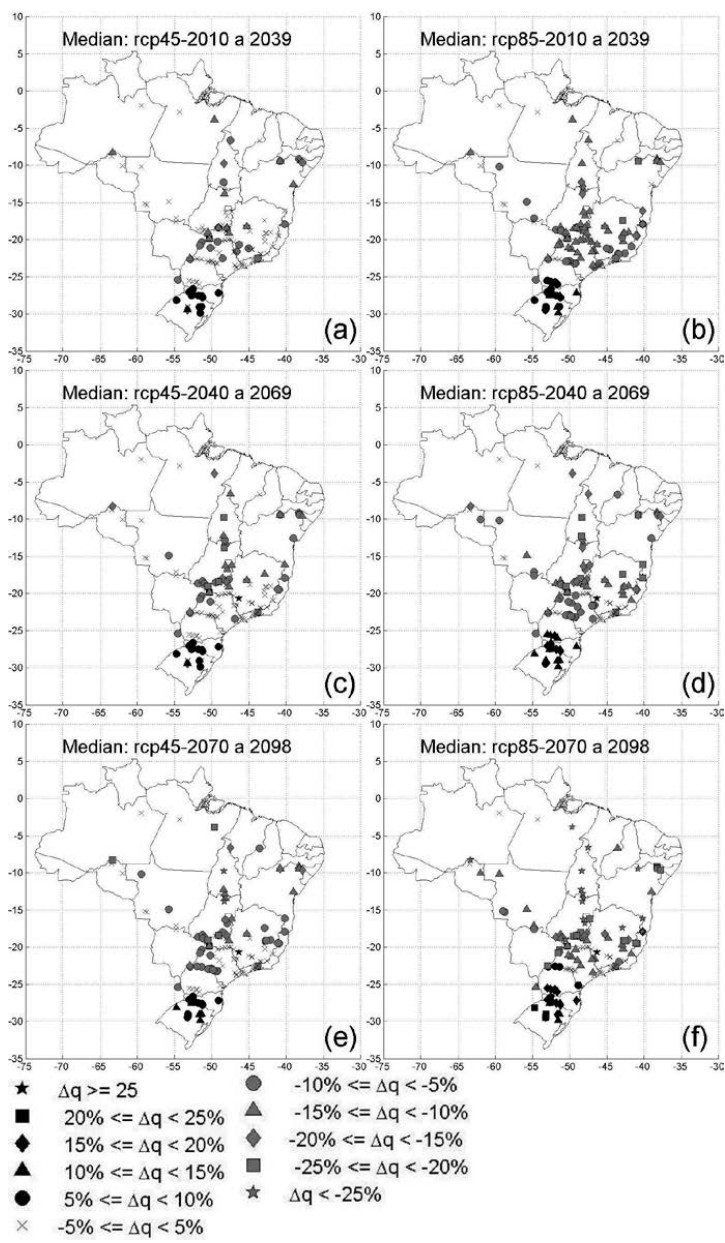

2    Figure 3 – Median of annual streamflow anomalies from IPCC-AR5 global models for 3 time-slices (2010-
3    2039, 2040-2069, and 2070-2098) under RCP4.5 and RCP8.5 scenarios.





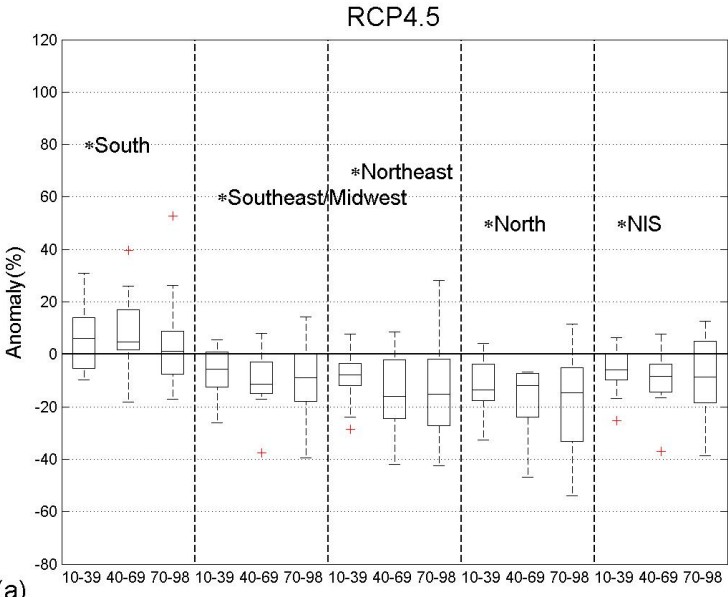

(a)

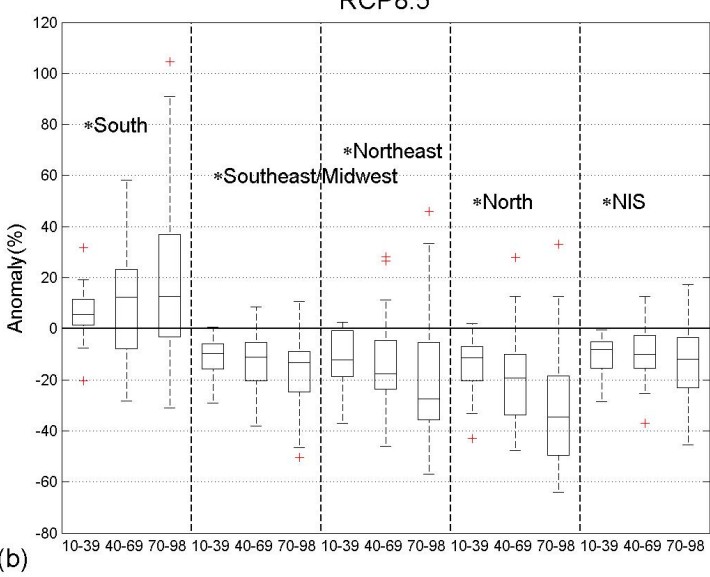

(b)

3 Figure 4 - ENA Annual Percentage Anomaly for the RCP 4.5 and RCP 8.5 scenario for the subsystem that
4 comprising the NIS. On axis X are assigned the three time-slices, 2010-2039, 2040-2069, and 2070-2098,
5 respectively for each group of boxplots.





Table 2 - Trend according to the Man-Kendall Sen method, in percentage per decade for the period 2010-2098.

| Models | SE | S | NE | N | NIS | SE | S | NE | N | NIS |
|---|---|---|---|---|---|---|---|---|---|---|
| BNU-ESM | -1,4 | - | -2,9 | -1,9 | -1,7 | -2,4 | - | -3,7 | - | -2,5 |
| CESM1-BGC | - | - | - | - | - | - | - | -3,9 | -2,8 | -1,6 |
| CSIRO-Mk3-6-0 | -2,1 | - | -2,6 | -3,7 | -2,1 | -2,8 | - | -3,7 | -4,0 | -2,8 |
| CanESM2 | -2,4 | - | -2,1 | -3,7 | -2,4 | -3,4 | - | -3,8 | -5,4 | -3,3 |
| HadGEM2-AO | 1,8 | 3,3 | - | - | 1,6 | - | 8,2 | 1,8 | -2,5 | - |
| IPSL-CM5A-LR | - | - | 2,3 | - | - | - | - | 5,5 | 3,2 | - |
| IPSL-CM5A-MR | 1,5 | - | 3,4 | 4,4 | 1,9 | 3,0 | - | 6,7 | 5,3 | 3,5 |
| MIROC-ESM_ | -2,3 | - | - | -2,9 | -1,8 | -6,2 | - | -1,9 | -5,4 | -5,1 |
| MIROC5 | - | - | - | - | - | - | 3,6 | -2,7 | - | - |
| bcc-csm1-1 | - | - | - | - | - | - | 2,3 | -1,6 | -3,0 | -1,1 |
| GFDL-ESM2M | * | * | * | * | * | -3,4 | -4,5 | -2,2 | -2,6 | -3,3 |
| GISS-E2-R | * | * | * | * | * | - | 5,0 | -3,6 | -2,9 | - |
| HadGEM2-CC | * | * | * | * | * | - | 13,2 | - | -6,1 | - |
| HadGEM2-ES | * | * | * | * | * | - | 10,6 | - | -5,2 | - |

*Model absence for the scenario
-Trend absence
