# Peer review of "Projections of the Affluent Natural Energy (ANE) for the"

_Hydrology and Earth System Sciences, 2016_

## Referee Comment (RC1) · Anonymous Referee #1 · 26 Apr 2016

This paper proposes to project the natural monthly water inflow to the 203 hydropower production schemes of the Brazilian National Interconnected System (including 97% of the Brazilian electricity production). The monthly inflows are simulated for 21 basins via a hydrologic model (SMAP) driven with monthly precipitation and potential evaporation estimates; for the remaining 182 basins, the monthly inflow is obtained via linear regression on the simulated 21 time series.

The topic as such (projection of streamflow for hydropower production) would be suitable for HESS. Being a scientific journal in the field of hydrology and earth system

sciences, the paper, and in particular its rather crude simulation method, can however not be considered suitable for publication. In fact, such an extremely simplified streamflow simulation method (direct use of global climate model output, linear regression between streamflow series) cannot be deemed useful to project climate change impacts.

---

## Referee Comment (RC2) · Anonymous Referee #2 · 2 May 2016

Silveira et al. present an assessment of Affluent Natural Energy (ANE) for the Brazilian National Interconnected System (NIS) using IPCC – AR5 projections according to RCP 4.5 and RCP 8.5. They calibrated a hydrologic model (SMAP) using historical (observed) data from 21 basins within NIS. Then, they used projected precipitation from a set of models to estimate streamflow in these 21 basins (using a monthly time step). To do this, an estimation of monthly evapotranspiration was calculated using the method by Penman-Monteith. Eventually, they estimated streamflow in all other stations of NIS (167) by using a stepwise regression calibrated on flows observed in the first 21 basins during the period 1931 – 2009. In this way, they propose a national-wide evaluation of

[Figure]

ANE for the entire NIS. This reveals that mean annual streamflow (hence, ANE) might decline for most basins considered. Spatial patterns of expected ANE show that this might nonetheless increase in southern areas of Brazil.

The assessment of future scenarios of hydroelectric power generation has been a relevant topic in modern hydrology since the seminal papers by, e.g., Christensen et al. 2004 (The Effects of Climate Change on the Hydrology and Water Resources of the Colorado River Basin, Climatic Change) or Lehner et al. 2005 (The impact of global change on the hydropower potential of Europe: a model-based analysis, Energy Policy). As a result, a substantial amount of papers discusses this topic in detail. However, outcomes of future projections usually show a remarkable spatial variability (see Schaefli 2015 on this, DOI 10.1002/wat2.1083). Furthermore, Brazil represents an interesting case study as hydropower plays a key role in its energetic budget, while its climate is spatially diverse and includes areas where ANE is strongly seasonal and other areas where this seasonality is lower. For all these reasons, I am convinced that the general topic of the manuscript is interesting and relevant for readers of HESS.

However, I think that the paper needs some substantial elaborations for publication in the journal. Some remarks are methodological, while some others regard the way this analysis is presented. I am confident that the manuscript would benefit from additional elaboration on both. In view of the following discussion, I recommend to re-consider this manuscript after major revisions. I would be happy to act as referee of a revised manuscript.

My first remark regards the method used to estimate ANE in the 167 basins/points where the model is not run (as already mentioned by Referee #1). Of course, I am aware that spatial modelling is sometimes cumbersome due to both, computational time and the lack of a proper characterization of the spatial variability of processes, parameters and input data. However, I do think that additional details and elaborations are needed on this point. First of all, authors should clearly discuss why they did not use the model in all basins considered. Is this because input or evaluation data are

missing? Is it because computational times are too high? This is a key point because, in the present version, the manuscript is rather vague on this point and readers lack a proper insight into the reasons of this choice (and the expected sensitivity of results to this assumption). If this is the result of a trade-off, this trade-off should be clearly introduced and discussed. Furthermore, stepwise regression of historical data is a very pragmatic choice that might trigger large uncertainty. I think simple solutions are appealing, but the manuscript lacks a proper discussion about, e.g., the performance of this regression for historical data, or reasons why these regressions are assumed valid for future climate. In the literature, a large amount of strategies is available to cope with the spatialization of input data or parameters (e.g., regionalization). I think that this analysis would clearly benefit from including the entire domain of the study in a unique simulation. According to my opinion, this methodological issue is the most important point to be addressed in the revised manuscript.

Another key issue of this manuscript is its lack of important details about all modelling steps. I will use Section 3.3 as an example, but I think this remark is very general and should be considered while revising the entire manuscript. When introducing the first step of the procedure, I missed key details about, e.g., the criteria used to select the 21 basins, reasons why these are considered "representative", the years used to calibrate the model, or the strategies implemented to evaluate it. Did you consider splitting the sample of observations in a calibration and evaluation period? How many years did you use during the calibration? How is model performance? How many years did you consider for evaluation? Which performance indicators did you consider? Similar remarks could be raised for all other steps of the procedure. For example, more details would be useful to understand and judge the bias removal procedure (how it works operationally?), the calculation of evapotranspiration (how did you set up the parameters of the Penman-Monteith method? How did you cope with spatial variability?), or the calibration of the regression used to regionalize the results (did you calculate an $R^2$ for this regression? Is the performance satisfactory? Do you have a concise equation that describes this regression and that might be reported in the text?).

Addressing all these points is very important as the paper often reads like a technical report, while papers should ensure repeatability and access to all the methods used by the authors.

Furthermore, the paper would clearly benefit from a thorough discussion of results. At present, the paper ends with a set of site-specific results (see Section 4) and with a concise conclusion, which nonetheless is very general. In between, authors should clearly contextualize their findings in the existing literature, they should identify advancements with respect to previous results, and they should tackle possible remarks. In a few words, they should present and discuss the implications of this work for a global scientific public. This is urgently needed to broaden the impact of the paper, which is now pretty limited.

SPECIFIC COMMENTS

1) The Affluent Natural Energy is sometimes called ANE (title), ENA (line 29) or NAE (abstract). Please choose one and use it consistently;

2) I found the Abstract well written. On the other hand, the Introduction is quite recursive: lines 3 – 8 page 2 introduce Climate Change. Then, the Brazilian case is defined (lines 9 – 22 page 2). But then, lines 23 page 2 – 6 page 3 deal again with Climate Change, while lines 6 – 26 page 3 re-focus on Brazil. I suggest rearranging.

3) Another important point that I missed in the Introduction is a proper identification of gaps and limitations of the state of the art. This is key as it should justify the motivation for an additional work on this topic. Please consider elaborating on this point.

4) I also suggest to report a formula for ANE/ENA/NAE in the introduction.

5) Section 2: please consider including an external reference (e.g., a web site) for readers interested in getting more information about NIS. I also suggest replacing lines 12 – 17 with a schematic of all regions and mutual fluxes.

6) Sections 3.1 and 3.2: I suggest merging these two sections. Please specify which

models in Table 1 did you consider and why.

7) Section 3.3: please consider my questions above.

8) Section 3.4: I did not find ONS 2010 in the reference list. I suppose that the j reported in Eq. 1 should be i, as this index is clearly indicated in the summation. Please specify the units for all the variables included in this Equation.

9) Section 4: how did you calculate the anomalies? Which is the control period? Could you please provide an equation for this? I think conclusions at lines 8 – 13 page 8 should be supported by further analysis. What is a "very expressive magnitude" (line 16)?

10) Section 5: I think that lines 7 – 16 page 9 are not supported by specific results. I think that they are inappropriate given the focus of this paper. They should be replaced with a more specific discussion of results and implications for a wider public.

In general, I found figures quite clear. I suggest enlarging captions and labels in Figure 1. Moreover, a coloured version of Figure 3 would probably be better. I also suggest a careful proofreading.

---

## Author Comment (AC1) · 4 Jun 2016

1. "The topic as such (projection of streamflow for hydropower production) would be suitable for HESS."

a. Ok. Thanks.

2. "rather crude simulation method" . . . "simplified streamflow simulation method (direct use of global climate model output, linear regression between streamflow series)"

a. We use global model directly because the size of the water basin is big enough to

cover plenty of pixels from the global models. b. We performed a statistical downscaling to correct the bias from the global models. c. Regionalization of the streamflow is a methodology using nowadays in the ONS. This procedure used linear regression. The reservoirs considered in base stations are representative for each hydrosystem.

3. "cannot be deemed useful to project climate change impacts"

a. We evaluate the methodology for twenty century and calibration for hydrological model is good, then the methodology could represent the rainfall-runoff transformation. b. The procedure to calculate the Affluent Natural Energy was based on methodology used by ONS. It is considered the best on operational level in Brazil. c. Therefore, we evaluate that is reliable to estimate the impacts of the Climate Change on Streamflow and Hydropower production in Brazil.

---

## Author Comment (AC2) · 8 Jun 2016

1. "My first remark regards the method used to estimate ANE in the 167 basins/points where the model is not run (as already mentioned by Referee #1)".

a. We used one or more stations to represent each one water basin and then we regionalize the streamflow for the other stations. These 167 basins are considered sub-basins or have a very close behavior of the representative station.

2. "First of all, authors should clearly discuss why they did not use the model in all basins considered. Is this because input or evaluation data are missing? Is it because

computational times are too high? ... why the linear regression are assume valid for future climate. "

a. They have the same atmospheric system and climatology.

b. Our hypothesis is the climate change modify intensity and frequency of the climate system occurrence, however it does not change the main rainfall system for each climate regions in Brazil.

c. In addition, see 1.a.

3. "In the literature, a large amount of strategies is available to cope with the spatialization of input data or parameters (e.g., regionalization). I think that this analysis would clearly benefit from including the entire domain of the study in a unique simulation. According to my opinion, this methodological issue is the most important point to be addressed in the revised manuscript."

a. The spatial structure is provided by the precipitation field from the climate change global models.

b. Entire domain was covered by the twenty-one reference stations. There is a high correlation among the reference and remain stations.

c. The uncertainty from the climate change global models is higher than streamflow regionalization uncertainty using regression modeling.

4. "Another key issue of this manuscript is its lack of important details about all modelling steps. ..."

a. We agreed with the referee and will incorporate the answers of all the questions in the revised manuscript.

5. "In a few words, they should present and discuss the implications of this work for a global scientific public."

a. Technology and Policy point. First of all, the results showed the projection strong reduction in hydropower production in Northern Brazil. The main investments on expansion of hydropower in Brazil are concentrated on Amazon region which lead to environmental/social impacts and vulnerabilities due to climate projections to this regions. Gas Natural/Coal-Fired thermoelectric power plants have been primarily alternative during shortage of hydropower to secure the supplies. Also from this work, achieving power and food security combined with sustainable development Brazil policies needs to be more proactive for promoting power mitigation plans and attaining alternative clean power sources.

b. Scientific and Methodology point This paper proposed a methodology to assess the climate change impacts on the climate-water-energy nexus in Brazil and thus show the potential problems or opportunities in the expansion of hydroelectric capacity. This proposed methodology can be used by other countries that have significant presence of hydroelectric power in the power grid or be updated after the increase in hydropower plants in Brazil.

6. Specific Comments: We agree with all specific comments and we will address them on the new manuscript version.